# Unveiling a Meaningful Form of *Cypripedium* × *ventricosum* Sw. (Cypripedioideae, Orchidaceae) from Changbai Mountain, China: Insights from Morphological, Molecular, and Plastome Analyses

**DOI:** 10.3390/plants14050772

**Published:** 2025-03-03

**Authors:** Ying Li, Xi Lu, Shuang Li, Yue Sun, Yuze Shan, Shizhuo Wang, Nan Jiang, Yiting Xiao, Qi Wang, Jiahui Yu, Qingtao Cao, Sulei Wu, Lifei Chen, Xinzhu Dai

**Affiliations:** 1College of Forestry and Grassland, Jilin Agricultural University, 2888 Xincheng Street, Changchun 130118, China; yingli1106@163.com (Y.L.); shuangli_037@163.com (S.L.); s595084179@163.com (Y.S.); w1085581514@163.com (S.W.); 13234407788@163.com (J.Y.); wsl18123462593@163.com (S.W.); 2College of Horticulture, Jilin Agricultural University, 2888 Xincheng Street, Changchun 130118, China; luxi@jlau.edu.cn (X.L.); 15524323842@163.com (Y.S.); jiangnan09160420@163.com (N.J.); xiaoyitingo_0@163.com (Y.X.); w3977362@163.com (Q.W.); 15961328355@163.com (Q.C.); 3Changchun Academy of Forestry, 5840 Jingyue Street, Jingyuetan Tourist Economic Development Zone, Changchun 130117, China

**Keywords:** *Cypripedium*, molecular phylogeny, backcross, evolution, chloroplast, extinction, conservation

## Abstract

A *Cypripedium* plant was discovered in Wangqing County, Changbai Mountain, Jilin Province. This newly discovered plant of *Cypripedium* not only inhabits the same natural habitat as *Cypripedium calceolus* L. and *Cypripedium* × *ventricosum* Sw. but also has a morphology intermediate between that of *C. calceolus* and *C.* × *ventricosum*. Its dorsal sepals, petals, and synsepal are similar to those of *C. calceolus*, while the color of its lip is intermediate between that of *C. calceolus* and *C.* × *ventricosum*. For the purpose of distinguishing the newly discovered plant of *Cypripedium* from other *Cypripedium* plants, we provisionally named it W1. To further verify the taxonomic status of W1, we introduced three identified forms of *C.* × *ventricosum* and conducted molecular biology analyses with W1, *C. calceolus*, and *C.* × *ventricosum*. The analyses further confirmed the relationship between W1 and *C.* × *ventricosum*, and the phylogenetic analysis of the nuclear region demonstrated a close relationship between W1 and *C. calceolus*. Collectively, the morphological and molecular evidence indicates that W1 is a product of the backcross between *C.* × *ventricosum* and *C. calceolus*. Although it shows morphological differences from typical *C.* × *ventricosum*, it can still be considered a form of *C.* × *ventricosum*. We further investigated the chloroplast genome of this form of *C.* × *ventricosum* and determined that its total genome length was 196,850 bp. The genome contains 132 genes, including 87 protein-coding genes, 37 tRNA genes, and 8 rRNA genes. By analyzing the phylogenetic position and chloroplast genome of the form of *C.* × *ventricosum*, this study clarified the relationships among *Cypripedium* taxa with similar morphological characteristics, laying a foundation for research on orchid evolution and species conservation.

## 1. Introduction

*Cypripedium* L., a genus within the Orchidaceae, which belongs to Cypripedioideae, encompasses around 50 species globally. They are mainly distributed in the temperate and subtropical mountainous regions of the Northern Hemisphere, including East Asia, North America, and Europe. Most species of *Cypripedium* are protected under the Convention on International Trade in Endangered Species of Wild Fauna and Flora (CITES) [1]. There are 36 species (more than half of the endemic species) of *Cypripedium* distributed in China, thereby establishing it as one of the diversity centers of *Cypripedium* [2,3].

A more comprehensive orchid phylogeny serves as the foundation for understanding the evolution and diversity of orchids, covering individual characteristics, evolutionary processes, and species diversity. *Cypripedium* species typically feature plicate leaves and unilocular ovaries with parietal placentation [4]. However, some scholars have questioned the reliability of these features [5,6]. During in-depth investigations of *Cypripedium*, numerous classification revisions have been produced. Lindley [7] described 22 species in the genus *Cypripedium* and further classified them into several subgeneric groups. Pfitzer considered the genus *Cypripedium* to comprise 28 species in four sections [8]. In Cribb’s taxonomic fragment, the number of species increased to 45, and the number of sections expanded to 11 [4]. Meanwhile, Eccarius reclassified eight species as subspecies or varieties and re-divided *Cypripedium* into 2 sub-genera, 13 sections, and 37 species [9].

The increase in molecular markers (such as plastid markers, nuclear ribosomal DNA, and mitochondrial or low-copy nuclear genes) has significantly advanced our understanding of relationships among orchid species and revealed the connections between orchid subfamilies with morphologically ambiguous taxa [10,11,12,13,14]. Recent molecular phylogenetic studies on *Cypripedium* (nuclear ribosomal ITS, low-copy nuclear gene, and chloroplast genes such as *matK*, *rbcL*, *trnH-psbA*, *atpI-atpH*, *trnS-trnfM*, and *trnL-F*) supported that *Cypripedium* was monophyletic and it was approximately divided into 15 sections [3,15,16]. Certain studies have utilized various phylogenetic reconstruction methodologies including maximum likelihood (ML) and Bayesian inference (BI) [17]. Most researchers used different combinations of up to eight Sanger-sequencing nuclear makers [18] and chloroplast DNA markers [16,19]. In contrast, the most recent studies have resorted to plastome data and 41 nuclear loci obtained from transcriptomes [20].

Hybridization is a frequent phenomenon among plant species in the wild and is regarded as a crucial driver of plant evolution [21,22]. Orchids have relatively weak genetic barriers to hybridization and highly evolved pollination systems. Approximately one-third of orchid species achieve pollination by deceiving insects. This pollination mechanism increases the probability of interspecific hybridization among genetically related, co-existing taxa with overlapping flowering periods [23,24,25,26,27]. Previous studies have identified and described natural hybrids of *Cypripedium* in regions with relatively simple vegetation and concentrated flowering periods and emphasized the occurrence and significance of hybridization in the diversity and evolution of *Cypripedium* [28,29]. For example, a *Cypripedium* species was discovered in Shangri-La, Yunnan. The colors of its sepals, petals, and lip are intermediate between those of *Cypripedium yunnanense* Franch. and *Cypripedium tibeticum* King ex Rolfe, leading to the speculation that it might be a hybrid of *C. yunnanense* and *C. tibeticum*. AFLP analysis confirmed that the “suspected” hybrid is indeed the product of hybridization between *C. yunnanense* and *C. tibeticum*. Hybridization occurred between the parents and between the F1 generation and *C. yunnanense*. Unidirectional gene introgression occurred from *C. tibeticum* to *C. yunnanense*, but no hybrid offspring were produced between the F1 generation and *C. yunnanense* [30,31,32].

Chloroplast genome sequences are widely employed in phylogenetic and divergence history studies of flowering plants. Compared with nuclear and mitochondrial genomes, they possess characteristics such as maternal inheritance, numerous variation sites, and sequence conservation [33,34]. Chloroplast genome sequences are widely used in genetic engineering, phylogenetics, species identification, and diversity research [35,36]. They contain sufficient genetic material for comparative analysis and species diversification studies, as they house functional genes vital for plant cells [37]. Moreover, chloroplast genomes are capable of providing distinctive and copious information for dissecting plant systematics and evolutionary associations on account of their matrilineal inheritance features [38]. Meanwhile, the highly variable loci in the chloroplast genome can greatly contribute to future phylogenetic studies of the genus [39].

During the field investigation, we discovered a *Cypripedium* plant and provisionally named it W1. Based on its morphology and habitat environment, we suspect that it might be a hybrid of *C.* × *ventricosum* and *C. calceolus*. To verify this hypothesis, we also introduced three forms found during the investigation and confirmed to be *C.* × *ventricosum* [15]. These forms are morphologically similar to *C.* × *ventricosum* but differ from W1. So, is W1 a natural hybrid or a form of *C.* × *ventricosum*? Therefore, we utilized morphological comparison and molecular biology techniques to identify *C.* × *ventricosum*, *C. calceolus*, and W1. In addition, three forms of *C.* × *ventricosum* discovered during the investigation were also included for comparative verification to determine the phylogenetic position of W1. Meanwhile, we conducted an in-depth exploration of the chloroplast genome structure of W1, aiming to comprehensively clarify the taxonomic status of W1.

## 2. Material and Methods

### 2.1. Plant Materials

The plant materials were collected from Wangqing County, Yanbian Chaoxianzu (Korean) Autonomous Prefecture, Jilin Province (Figure 1). The collected plant materials included *Cypripedium* × *ventricosum*, *C. calceolus*, and newly discovered plants of *Cypripedium* from the same taxonomic group. Meanwhile, to validate the relationships among newly discovered plants of *Cypripedium*, *C.* × *ventricosum*, and *C. calceolus*, 3 forms of *C.* × *ventricosum* discovered during the investigation were selected as references. All 3 forms of *C.* × *ventricosum* have been authenticated. The newly discovered plants of *Cypripedium* and 3 forms of *C.* × *ventricosum* from different taxonomic groups were tentatively named W1, W2, W3, and W4, respectively. An amount of 1 g of leaves was taken from *C. × ventricosum*, *C. calceolus,* and W1, W2, W3, and W4 separately. Leaves were selected from young individuals with vigorous growth and free from pests and diseases. Subsequently, the leaves were desiccated using silica gel for preservation.

### 2.2. Morphological Analysis

Images of flowering plants and floral anatomy were photographed using a Canon camera (Canon Inc., EOS M3 Mini SLR, Ota City, Japan). We utilized a measuring tape and vernier calipers to quantify the morphological traits of W1, *C.* × *ventricosum*, *C. calceolus,* W2, W3, and W4. The colors of the flower parts of *C. × ventricosum*, *C. calceolus*, W1, W2, W3, and W4 were compared with the RHS Colour Chart (Sixth Edition: 2019 reprint Published by RHS Media, Royal Horticultural Society, 80 Vincent Square, London SW1P 2PE, UK).

### 2.3. Extraction and Sequencing of Chloroplast Genomic DNA

We dried the leaves with silica gel and applied the modified CTAB protocol to extract the DNA of *Cypripedium* × *ventricosum*, *C. calceolus*, W1, W2, W3, and W4 [40]. The extracted genomic DNA was fragmented into 150 bp segments, and sequencing libraries were constructed using the TruSeq DNA Sample LT Prep kit (Illumina, San Diego, CA, USA) following the manufacturer’s instructions. Subsequently, these libraries were subjected to sequencing on the Illumina NovaSeq platform. The sequencing was conducted by Guangzhou Furuikanghe Biotechnology Co., Ltd. (Guangzhou, China).

### 2.4. Assembly and Annotation of Chloroplast Genome Sequence

To obtain complete plastomes, we used a custom script with default parameters (-L 5, -p 0.5, -N 0.1) to filter the raw sequencing data, removing low-quality or noisy reads. Paired reads were discarded when the N content surpassed 10% of the read base number or when the count of low-quality (Q ≤ 5) bases accounted for more than 50% of the read. Next, we processed the filtered paired-end reads [41] using GetOrganelle v1.7.7.0 with default parameters [19]. Plastome annotation was performed using DOGMA v33.1 [42]. This tool utilizes the BLAST (https://blast.ncbi.nlm.nih.gov/Blast.cgi accessed on 9 June 2024) search function to identify various genes (including protein-coding genes, tRNA, and rRNA-coding genes) through homologous alignment with published plastome data of Orchidaceae. We manually aligned and confirmed the plastome sequences with Geneious v11.1.5 [43], comparing them with previously published *Cypripedium* plastomes. Subsequently, we visualized the circular genome maps using OGDRAW (https://chlorobox.mpimp-golm.mpg.de/OGDraw.html accessed on 9 June 2024).

### 2.5. Codon Usage Bias

For the codon usage bias analysis, the protein-coding sequences from the chloroplast genome of W1 were screened with Geneious 11.1.5 [43] and MEGA 11.0.1 [44]. The codon preference index can be affected by sequence repetitions shorter than 300 bp. After removing these repetitive sequences, we retrieved sequences with ATG start codons and TAG, TGA, or TAA stop codons for further analysis. Ultimately, 51 coding sequences (CDS) were obtained and served as sample sequences for the study. CodonW 1.4.2 and EMBOSS (http://imed.med.ucm.es/Tools/EMBOSS accessed on 9 June 2024) were utilized to examine the Relative Synonymous Codon Usage (RSCU). We calculated the number of nucleotides A, T, C, and G at the third codon position (denoted as A3, T3, C3, and G3, respectively), the Frequency of Optimal Codons (FOP), the Codon Adaptation Index (CAI), the Codon Bias Index (CBI), and the Effective Number of Codons (ENC). Finally, we generated synonymous RSCU maps using the R language.

### 2.6. Analysis of Repeat Sequences

We utilized MISA (https://webblast.ipk-gatersleben.de/misa/ accessed on 10 June 2024) for the accurate annotation of repetitive sequences in the chloroplast genome of W1. The minimum thresholds for simple repetitive sequences of mononucleotides, dinucleotides, trinucleotides, tetranucleotides, pentanucleotides, and hexanucleotides were set to 10, 6, 5, 5, 5, and 5 respectively. Reputer (https://bibiserv.cebitec.uni-bielefeld.de/reputer accessed on 10 June 2024) was used to assess forward, palindromic, reverse, and complement sequence repeats.

### 2.7. Comparative Genome Analysis

To gain a more in-depth comprehension of the structural and compositional characteristics of the *Cypripedium* chloroplast genome, we used R language and IRscope to conduct a comparative mapping analysis of the chloroplast genomes between W1 and 9 closely related *Cypripedium* species.

### 2.8. Phylogenetic Analysis

In this analysis, we used 5 markers: nuclear ribosomal ITS DNA, *trnH-psbA*, *trnS-trnfM*, *trnL* intron, and *trnL-F* spacer. Voucher specimen information and GenBank accession numbers are presented in Table 1. A total of 37 *Cypripedium* accessions were incorporated into the phylogenetic analysis, representing plants from 8 sections of *Cypripedium*. Based on a prior study [45], 4 species, *Mexipedium xerophyticum* (Soto Arenas, Salazar & Hágsater) V. A. Albert & M. W. Chase, *Paphiopedilum malipoense* S. C. Chen & Z. H. Tsi, *Phragmipedium exstaminodium* (Castaño, Hágsater & E. Aguirre) P. J. Cribb & Purver, and *Phrag. schlimii* (Linden & Rchb. f.) Rolfe, were selected as an outgroup. The primers utilized were identical to those previously employed by Li et al. [3]. Based on nuclear gene sequences and chloroplast data, we established 2 matrices and then examined them sequentially to construct a combined matrix. For phylogenetic analysis, MEGA11.0.1 was used to construct phylogenetic trees with the Maximum Parsimony (MP) and Maximum Likelihood (ML) methods, aiming to infer the evolutionary relationships among species. A heuristic search approach was selected to find the optimal phylogenetic tree. Specifically, the Tree Bisection–Reconnection (TBR) algorithm was utilized for the search and optimization of the tree topology. To evaluate the reliability of each branch in both the MP and ML trees, the Bootstrap Method was applied. The number of bootstrap replications was set to 1000. In each replication, sequences of the same length as the original data were randomly sampled with replacements from the original sequence dataset to construct new datasets. Subsequently, phylogenetic trees were reconstructed based on these new datasets.

## 3. Results

### 3.1. Morphological Characteristics

Figure 2 depicts the habitat of W1 and shows the comparison pictures between the flowers of W1, *C. × ventricosum*, and *C. calceolus*. By comparing the colors of the flower structures and petal shapes of W1 with those of *C. × ventricosum* and *C. calceolus*, we discerned the genetic patterns related to color and morphology. To provide a clearer view of the internal structure of W1 flowers, we show the dissected parts of its flower against a black background. The components are labeled F, G, H, I, and J and are symmetrically arranged in Figure 2. To clearly illustrate the internal structure of W1, a line drawing of W1 is provided concurrently (Figure 3).

To make a more precise comparison of the morphological characteristics between W1 and three forms of *C.* × *ventricosum*, the anatomical diagrams of the flowers of W2, W3, and W4 are presented in Figure 4. Although there are slight differences in color among W2, W3, and W4, the shapes of their sepals, dorsal sepals, and petals do not vary significantly. In contrast, the sepals, dorsal sepals, and petals of W1 are more similar to those of *C. calceolus*.

Table 2 presents a comparison of the morphological characteristics of W1, *C.* × *ventricosum*, *C. calceolus*, W2, W3, and W4. The colors of different flower parts were named according to the RHS Colour Chart in Table 3. The tip of the lip was a vivid greenish-yellow hue, gradually transitioning to greyish-brown. The dorsal sepal and synsepal showed a color gradient from bright greenish-yellow to deep red, while the petals were dark red.

### 3.2. Characteristics of Plastome of W1

The chloroplast genome of W1 displayed a typical double-stranded circular quadripartite structure, consisting of four distinct regions: a large single-copy region (LSC), a small single-copy region (SSC), and two inverted repeat regions. The total length of the genome was 196,850 bp, with a GC content of 31% and an AT content of 69% (Figure 5), indicating a significant AT bias. Notably, the GC content in LSC (27.2%) and SSC (21.4%) was considerably lower than that in IR regions (42.1%). Table 4 details the genes within the chloroplast genome of W1. It was observed that 20 genes existed in two copies, comprising four rRNAs, right tRNAs, and seven protein-coding genes. In addition, 12 genes contained one intron and 3 genes contained two introns.

### 3.3. Codon Usage Bias

Codon usage bias was calculated with the sequences of 51 protein-coding genes. These genes comprised a total of 21,260 codons. Cys had the lowest codon count (67, 0.32%) and Ile had the highest (881, 4.14%) (Table 5). Based on RSCU values, Figure 6 shows 33 codons with a high usage preference (RSCU > 1). Among these, four codons (UCC, UUG, UGG, and AUG) did not end with A or U. This indicated that protein-coding genes prefer A/U codon usage, reflecting a codon usage bias.

The average GCall content was 38.25%. The average GC content at the first, second, and third codon positions was 46.59%, 39.52%, and 28.93%, respectively (Figure 7). The codons of W1 favor A/T bases. The variation in GC content across different codon positions and the lower average content of the third base compared to the first two averages support this preference. Typically, an ENC (Effective Number of Codons) value of 45 serves as a benchmark for assessing the degree of gene codon usage bias. The ENC (Effective Number of Codons) value of W1 ranged from 37.92 to 60.61, with an average of 47.88. Among the genes, 37 genes had an ENC value greater than 45. Therefore, we can infer that the chloroplast genome of W1 displays a weak codon usage bias.

No significant association was found between GC3 vs. GC1 or GC2. However, a significant correlation existed between GC1 and GC2. Meanwhile, significant correlations were presented between GCall vs. GC1 and GC2 vs. GC3 (*p* < 0.01) (Table 6). Conversely, no significant relationship was observed between ENC vs. GC1, GC2, or GCall (*p* > 0.05). This implies that the content of the third base in W1 codons affects the codon usage preference to a certain extent.

Most of the chloroplast coding genes of W1 were below the standard curve, indicating a significant disparity between ENCobs and ENCexp for most of these genes. This suggests that natural selection had a greater impact on the chloroplast genome of W1 compared to the impact of mutation (Figure 8A). Neutral plot analysis, which illustrated the relationship between GC12 and GC3, further clarified how mutation pressure and natural selection affect codon bias. The regression coefficient revealed that natural selection accounted for 78.35% of the influence of variation pressure on codon usage, while mutation pressure contributed 21.6%. Consistent with ENC-plot analysis, natural selection had a more pronounced influence on the codon use bias of the W1 chloroplast genome than mutation pressure (Figure 8B). Under natural selection pressure, the usage of A/T and G/C was unequal. In contrast, when only mutation pressure was in effect, the unpredictability of mutation made the probabilities of the third codon base being A/T or C/G equal. The chloroplast coding genes of W1 were randomly distributed across four locations, mostly distant from the center. Additionally, most of the genes were located in the lower right corner. This indicates that the third codon preferred using T/G bases, with the usage frequencies following the order T>A and G>C (Figure 8C).

### 3.4. Repeat Sequence Analysis

In the chloroplast genome of W1, the vast majority of simple sequence repeats (SSRs) were composed solely of A or T, reflecting the total AT abundance of the complete chloroplast sequence (Figure 9A). Within the LSC region, the greatest number of SSRs (n = 91, 56.50%) was detected. In contrast, IR and SSC had 34 (21.10%) and 36 (22.40%), respectively (Figure 9B). Moreover, a total of 42 tandem repeats and 39 scattered repeats were identified in the chloroplast genome. Among the scattered repetitive sequences, there were 15 palindromic, 8 forward, 22 reverse, and 4 complementing repetitive sequences (Figure 9C).

### 3.5. Comparative Genome Analysis

As shown in Figure 10, the length of the IR regions of the 10 *Cypripedium* ranged from 27,524 to 28,079 bp. The *rpl22* was positioned on the left side of the LSC-IRb border, except for those in *Cypripedium farreri* W. W. Sm. and *C. yunnanense*, where *rps19* and *rpl22* were both distributed on the sides of the boundary. The *rps19* was situated on the right side of the LSC-IRb border. Similarly, the *rpl22* of the other eight *Cypripedium* genes was positioned at the LSC-IRb boundary. Regarding the *ndhF* genes, they were all located on the IRb-SSC boundary, except for those in *Cypripedium fasciolatum* Franch. and *Cypripedium shanxiense* S. C. Chen, which were located on the right side of the boundary. Notably, *ycf1* was absent from the IRb-SSC boundary in *C. shanxiense* and W1. In contrast, it was present in the IRb-SSC region of the other eight *Cypripedium* plants and overlapped with *ndhF*. The overlapping portion ranged from 28 to 68 bp. The *ndhF* was 28–68 bp in the IRb region and 2170–2192 bp in the SSC region. The *ycf1* was 1461–1469 bp in the IRb region and 1–29 bp in the SSC region. When positioned at the SSC-IRa boundary, the *ycf1* exhibited a length of 1462–2083 bp in the IRa region and 3671–4283 bp in the SSC region. Both the *rps19* and *psbA* were located on either side of the IRa-LSC border. Specifically, *rps19* in *Cypripedium macranthos* Sw., *C. farreri*, and *C. × ventricosum* was on the IRa-LSC boundary. In the SSC region, the *ycf1* gene had a length ranging from 1 to 29 bp, while in the IRb region, its length was 1461–1469 bp. Located on the SSC-IRa boundary, *ycf1* had a length of 1462–2083 bp in the IRa region and 3671–4283 bp in the SSC region. Additionally, two genes, *rps19,* and *psbA* were positioned on opposite sides of the IRa-LSC boundary (Figure 10).

### 3.6. Phylogenetic Analysis

The results of the statistical analysis of sequence information and phylogenetic trees are shown in Table 7. The combined matrix of nrITS and plastid regions was 4482 bp, containing 1789 (39.92%) variable sites and 1148 (25.61%) parsimony informative sites, Among the three matrices, nrITS has the highest proportion of variable sites and informative sites, thus providing more phylogenetic information than plastids. However, the plastid phylogenetic tree has higher consistency and retention indices, and its topological structure is more reliable. Phylogenetic relationships from ML and MP (Figure 11) were highly consistent with the total evidence trees. The phylogeny of the plastid region further validated the relationship of W1 to *C.* × *ventricosum*, while the nuclear region showed the close affinity of W1 with *C. calceolus*. This difference in phylogenetic signals between the nuclear and plastid regions strongly indicates that the newly discovered W1 is a form of *C.* × *ventricosum*, which is consistent with the morphological inferences.

## 4. Discussion

### 4.1. Morphological and Molecular Evidence of W1

*Cypripedium* has high ornamental value, so it is very necessary to deeply explore its classification and evolution. Our research has found that W1, *C.* × *ventricosum* and *C. calceolus* coexist in a natural habitat. Moreover, the morphology of W1 shows similarities to that of the other two *Cypripedium* species. The morphological analysis of W1 indicates that its dorsal sepal, petals, and synsepal are similar to those of *C. calceolus*, while the color of its lip is intermediate between that of *C.* × *ventricosum* and *C. calceolus*. When comparing the morphologies of the three forms of *C.* × *ventricosum* with those of W1, *C.* × *ventricosum*, and *C. calceolus*, it can be observed that although the colors of these three forms of *C.* × *ventricosum* have slight differences compared with *C.* × *ventricosum*, there are no obvious changes in structures such as the dorsal sepal, petals, and synsepal. However, some morphological indicators of W1 are significantly different from those of the three forms that have been confirmed as *C.* × *ventricosum*.

With these questions, we conducted molecular biology analyses. Molecular analyses revealed that *C.* × *ventricosum*, *C. calceolus*, and W1 are closely related. The topologies of the phylogenetic trees obtained from the nrITS (nuclear ribosomal internal transcribed spacer) and plastid (chloroplast and other genomes) datasets are inconsistent in the Maximum Likelihood (ML)/Maximum Parsimony (MP) analyses. The phylogenetic analysis based on the combined matrix strongly supports that W1 is the sister species of *C.* × *ventricosum* (BSML = 85, BSMP = 71). The phylogeny based on the plastid region further confirms W1’s relationship with *C.* × *ventricosum*, while the analysis based on the nuclear region supports the close relationship between W1 and *C. calceolus*. Combined morphological and molecular evidence indicates that although morphologically W1 is not consistent with *C.* × *ventricosum* and three forms of *C.* × *ventricosum*, W1 is a product of the backcross between *C.* × *ventricosum* and *C. calceolus*. According to Chapter H4.1 of the ICN (International Code of Nomenclature for algae, fungi, and plants) [46], when all the parental taxa can be hypothesized or are known, a hybrid taxon is defined to encompass all individuals resulting from the cross between representatives of the specified parental taxa (not limited to the F1 generation but also subsequent filial generations, as well as backcrosses and combinations thereof). Therefore, W1 is a form of *C.* × *ventricosum*.

Previous studies have shown that hybridization can enhance gene flow among populations. Through genetic recombination, some favorable genes are formed, which in turn increases the genetic diversity of populations. This may lead to the formation of reproductive isolation between parental species and hybrids, ultimately giving rise to new species and promoting global biodiversity [47,48]. However, backcrossing between hybrids and their parents may result in introgression. Rare species and isolated small populations may face genetic assimilation due to introgression, potentially leading to their extinction [49]. Currently, *Cypripedium* is in an endangered state in the existing ecological environment, facing the risk of extinction. Therefore, it is extremely urgent to conduct in-depth research on the endangered mechanisms of *Cypripedium* and take effective conservation measures.

### 4.2. Chloroplast Genome Characterization of W1

The chloroplast genome of orchid species predominantly consists of circular closed DNA and exists in multiple copies. It can be divided into four main regions. Two inverted repeat sequences separate the large single-copy region from the small single-copy region. Additionally, the chloroplast genome of *Cypripedium* is larger than the average terrestrial plant genome (151 kb) [50]. In this study, the chloroplast genome of W1 exhibited a remarkably high AT content, while the GC content was relatively low, which is not common in the chloroplast genome sequences of terrestrial plants [51]. Moreover, the GC content in the LSC and SSC regions (27.2% and 21.4%) was significantly lower than that in the IR region (42.1%), attributable to the reduction in AT nucleotides in four rRNA genes (*rrn23*, *rrn16*, *rrn4.5*, and *rrn5*) [52,53]. The codon usage pattern represents a fundamental genetic characteristic of organisms and is intricately linked to evolution, natural selection, and mutation [54]. Our research findings indicate that cysteine (Cys) was the least abundant amino acid, which aligns with the results reported in other angiosperm taxa [55]. The presence of repetitive sequences such as poly (A), poly (T), or poly (AT) regions within a noncoding segment of the single copy region, especially in the LSC region, contributed to the high AT content [56]. Analysis of the basic repetitive sequences in W1 revealed a total of 161 SSRs. These repetitive sequences have been extensively utilized in numerous studies focusing on genetic diversity, population dynamics, and the identification of closely related species [57,58]. A multitude of investigations have demonstrated that one of the primary mechanisms driving the variation in chloroplast genome size is the expansion and contraction of the IR region [59,60,61]. We analyzed the chloroplast genomes of 10 *Cypripedium*. Although the SSC and LSC regions were relatively conserved, the border genes exhibited varying degrees of movement.

## 5. Conclusions

In the research field of plants in *Cypripedium*, their complex reproductive characteristics have always been the focus of academic attention. For *Cypripedium* with backcrosses and natural hybrids, the delimitation of species and infraspecific plants is of great significance. In this study, through a comprehensive analysis of morphological and molecular characteristics, a form of *C.* × *ventricosum* was identified. Although W1 differs morphologically from *C.* × *ventricosum*, the combined analysis of its plastid and nuclear genomes, as well as the analysis of the chloroplast genome, has verified the genetic relationship between W1 and *C.* × *ventricosum* and further proves that the plants of *Cypripedium* generated through backcrossing within *C.* × *ventricosum* still belong to *C.* × *ventricosum*. This research not only enriches the genomic resources of *Cypripedium* but also clarifies the taxonomic boundaries of *C.* × *ventricosum*.

Frequent backcrossing and natural hybridization have rendered the genetic backgrounds of *Cypripedium* extremely complex. Precisely defining the species and subspecies of *Cypripedium* not only helps to deeply understand its evolutionary pathways but also enables us, based on the clear taxonomic status, to formulate more targeted conservation strategies. This can effectively protect genetic diversity and avoid improper conservation measures caused by taxonomic confusion. Future research will concentrate on evaluating the status of *Cypripedium* plant resources. This involves systematically assessing their distribution, habitat, abundance, growth patterns, and reproduction strategies. Additionally, in-depth basic ecological and population biological studies will be conducted. These efforts aim to provide a theoretical basis for formulating rational and targeted conservation strategies.

## Figures and Tables

**Figure 1 plants-14-00772-f001:**
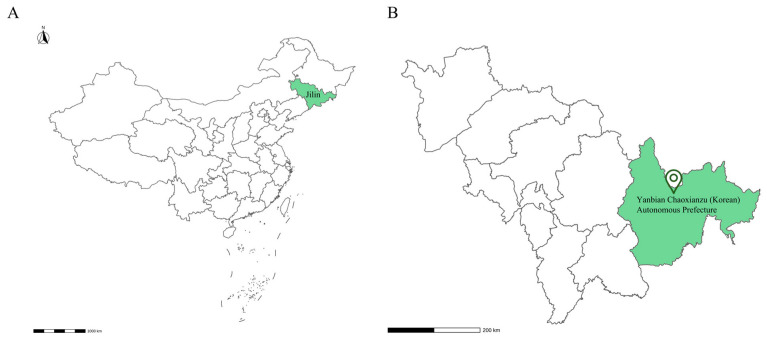
Location of W1, W2, W3, and W4. (**A**) China map. (**B**) Jilin Province.

**Figure 2 plants-14-00772-f002:**
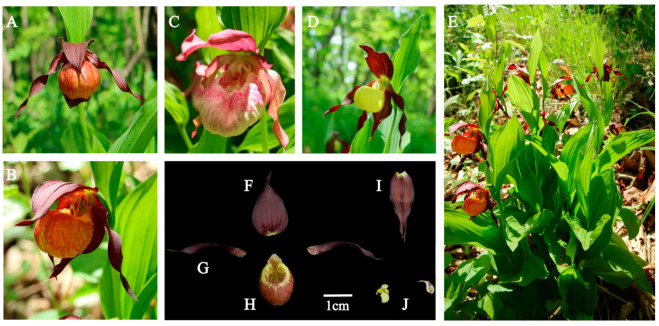
W1, *C.* × *ventricosum*, and *C. calceolus*. (**A**) The flower of W1 (front view). (**B**) The flower of W1 (lateral view). (**C**) *C.* × *ventricosum.* (**D**) *C. calceolus.* (**E**) The habitat and plant of W1. (**F**) The dorsal sepal of W1. (**G**) The petal of W1. (**H**) The lip of W1. (**I**) The synsepal of W1. (**J**) The column and stamonode of W1.

**Figure 3 plants-14-00772-f003:**
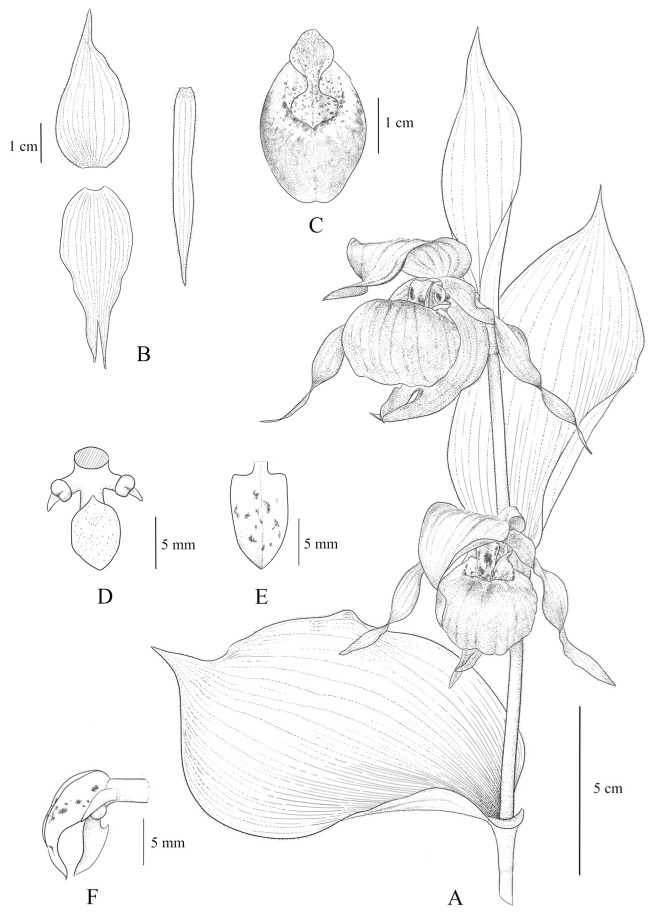
The line drawing of W1. (**A**) Flowering plant. (**B**) Dorsal sepal, synsepal, and petal. (**C**) Lip. (**D**) Column (front view). (**E**) Stamonode. (**F**) Column (lateral view).

**Figure 4 plants-14-00772-f004:**
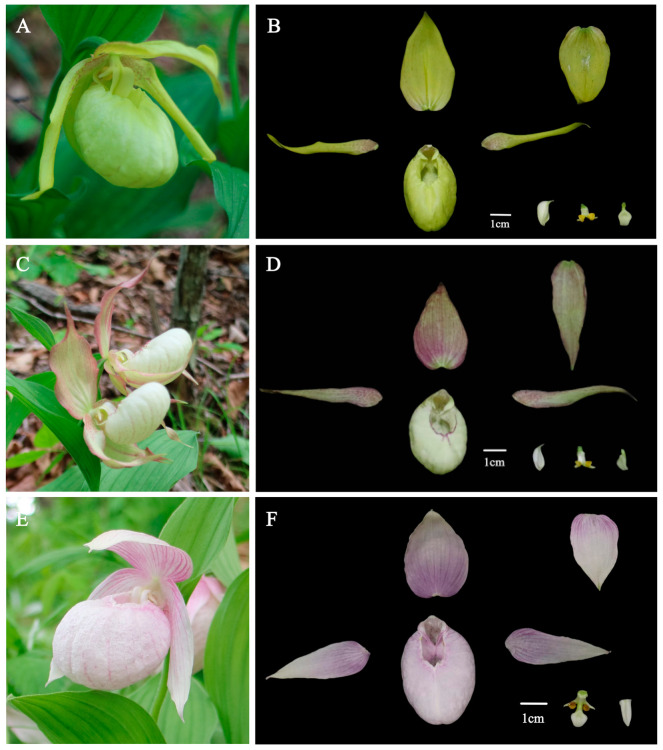
Three forms of *Cypripedium* × *ventricosum* and their anatomical diagrams. (**A**) The flower of W2. (**B**) The anatomical diagrams of W2. (**C**) The flower of W3. (**D**) The anatomical diagrams of W3. (**E**) The flower of W4. (**F**) The anatomical diagrams of W4.

**Figure 5 plants-14-00772-f005:**
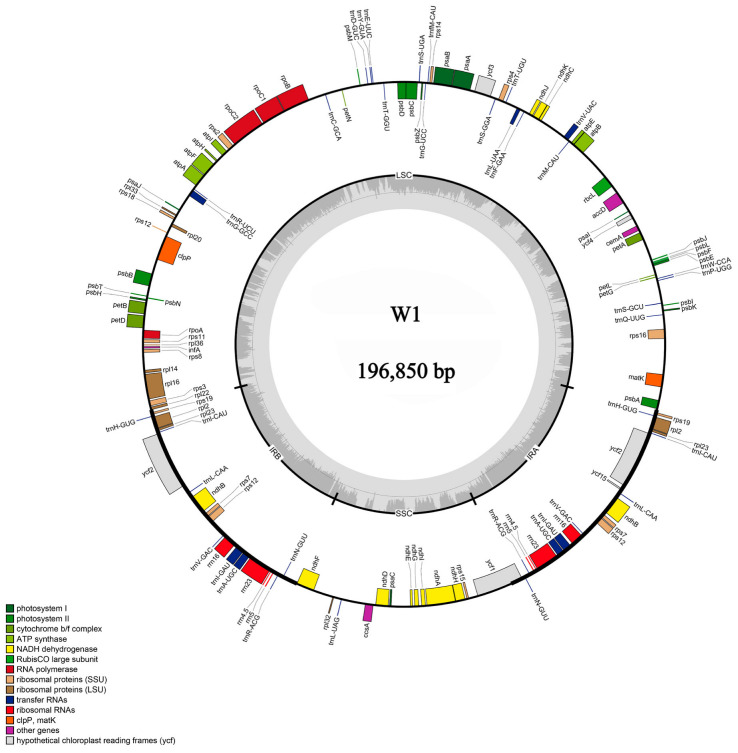
Chloroplast genome of W1.

**Figure 6 plants-14-00772-f006:**
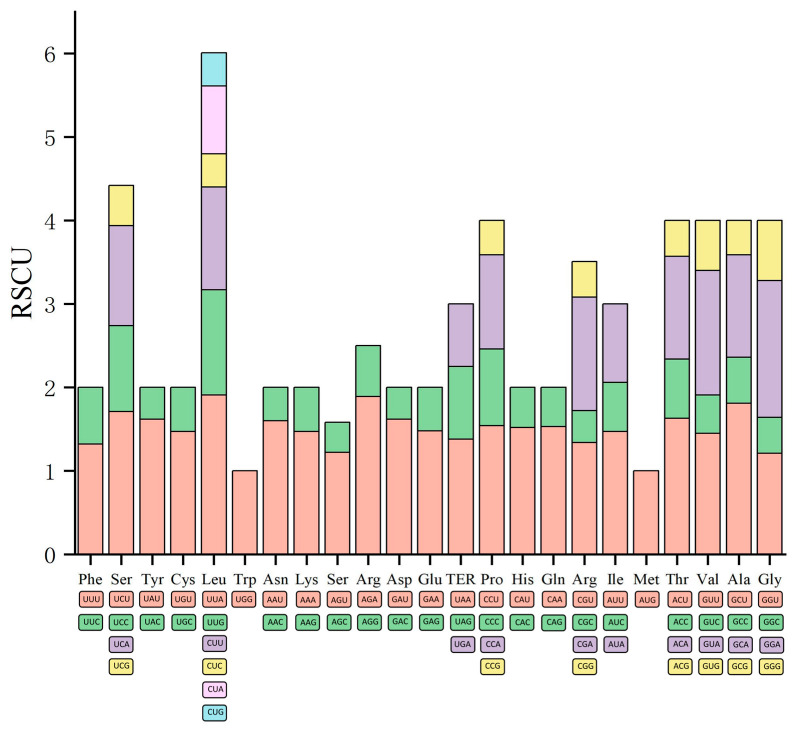
Relative synonymous codon usage (RSCU) of 23 amino acids in all protein-coding genes of the W1 chloroplast genome.

**Figure 7 plants-14-00772-f007:**
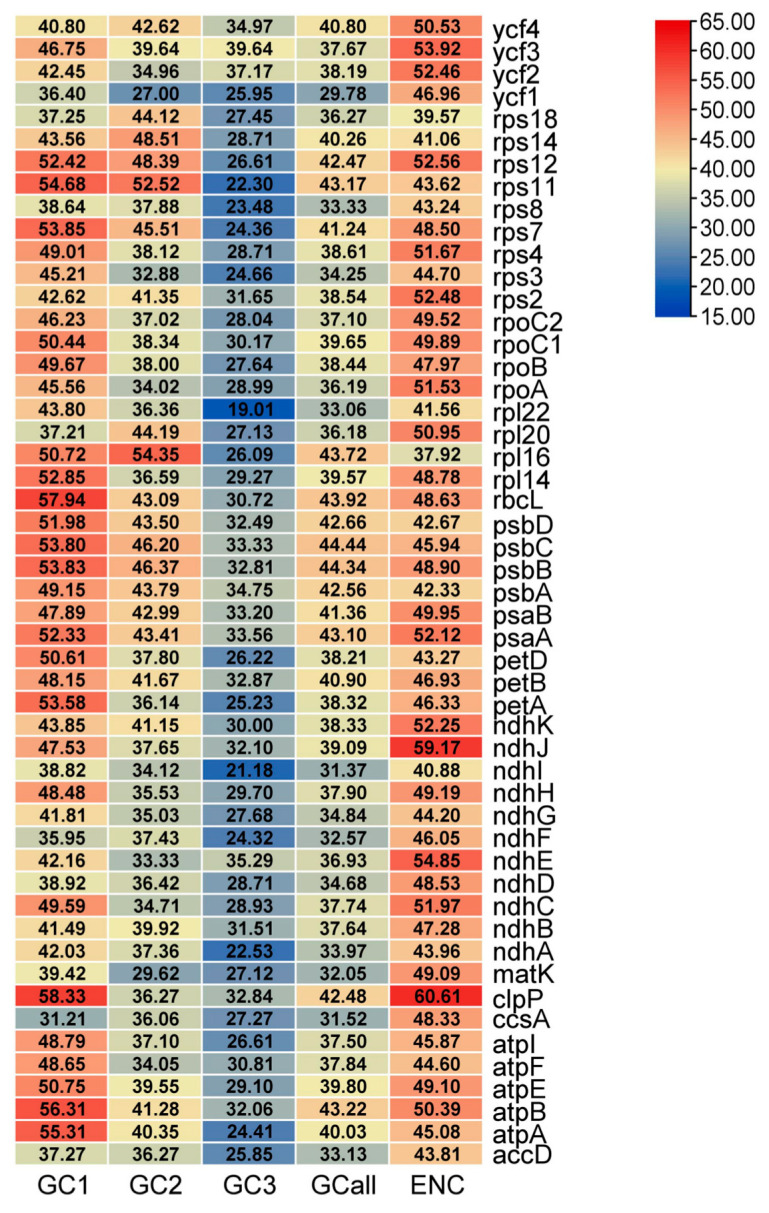
The GC content and ENC values of the chloroplast gene codon in W1.

**Figure 8 plants-14-00772-f008:**
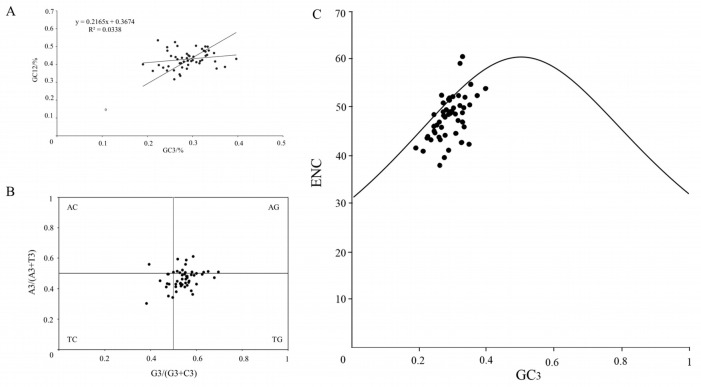
(**A**) Neutrality plot analysis. (**B**) PR2-plot analysis. (**C**) ENC-plot analysis.

**Figure 9 plants-14-00772-f009:**
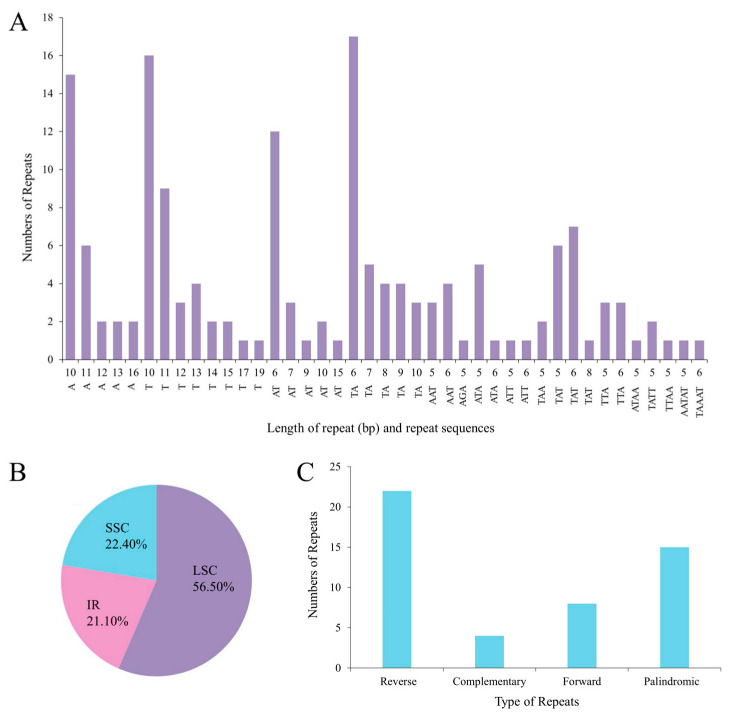
Repeat sequences and SSR analysis of the W1 chloroplast genome. (**A**) The number and length of each SSR. (**B**) Distribution of SSRs in the LSC, SSC, and IR regions of the W1 chloroplast genome. (**C**) The type and number of repeat sequences other than SSRs.

**Figure 10 plants-14-00772-f010:**
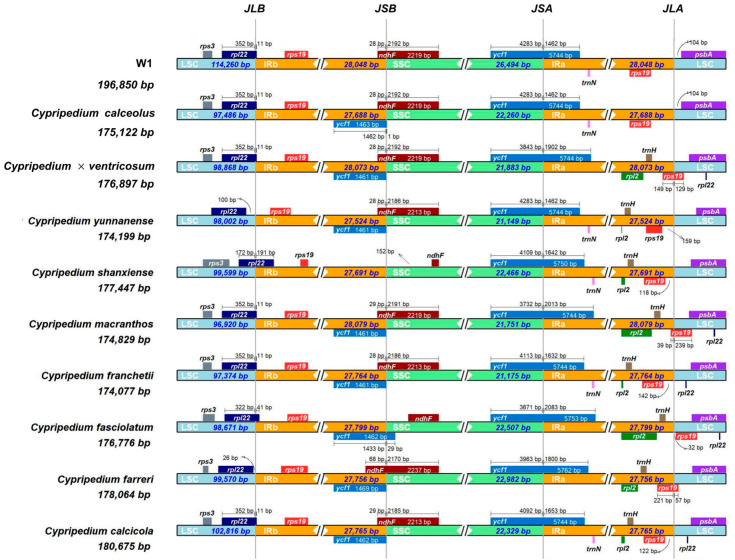
Comparison of the borders of the LSC, SSC, IRa (equal to IRA), and IRb (equal to IRB) regions between 10 species of *Cypripedium.*

**Figure 11 plants-14-00772-f011:**
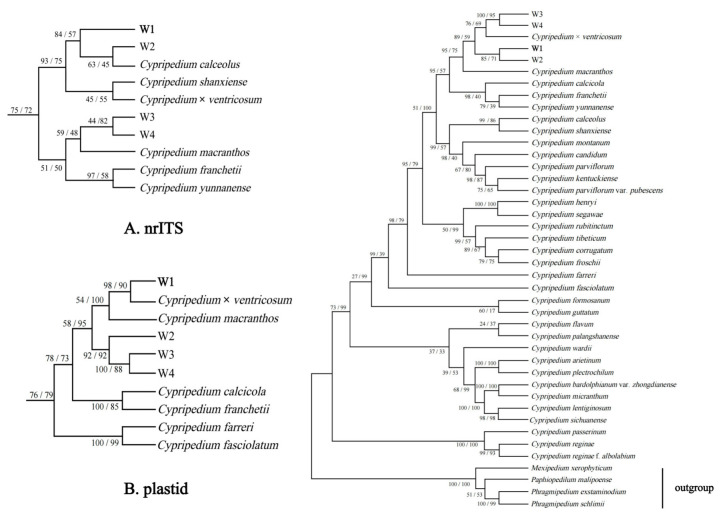
A maximum-likelihood analysis of the combined matrix obtained a phylogenetic tree. (**A**) nrITS, (**B**) plastid.

**Table 1 plants-14-00772-t001:** GenBank accession numbers for sequence data, and an asterisk (*) denotes sequences obtained in this study.

Taxa	Voucher Specimen	ITS	*trnH-psbA*	*trnS-trnfM*	*trnL* *Intron*	*trnL-F* Spacer
*C.* × *ventricosum*	-(NOCC)	JF796923	JF796979	JF797097	JF796882	JF797031
W1	CLF02	PQ619134	*	*	*	*
*C. bardolphianum*	Li JH S6163 (NOCC)	JF796900	JF796939	JF797058	-	JF797002
*C. calceolus*	Zhang Y E8044a	Z78521	JF796982	JF797102	JF796869	AY557224
*C. calcicola*	Li JH S6153 (NOCC)	JF796921	JF796977	JF797095	JF796880	JF797029
*C. candidum*	Steele B S1299a	EF370092	JF796942	JF797061	JF796864	JF797014
*C. corrugatum*	Gunnlaugson D S1149a	JF796901	JF796943	JF797062	JF796871	JF797020
*C. debile*	Zhong SW S6527 (PE)	JF796929	JF796987	JF797107	-	JF797048
*C. farreri*	Li JH S6181 (NOCC)	JF796904	JF796946	JF797064	JF796873	JF797022
*C. fasciculatum*	Kouzes R N6160 (PE)	JF796927	JF796984	JF797104	-	JF797044
*C. fasciolatum*	Li JH S6156 (NOCC)	Z78530	JF796947	JF797065	JF796874	JF797023
*C. flavum*	Li JH S6161 (NOCC)	Z78517	JF796948	JF797066	JF796857	JF797005
*C. formosanum*	Gunnlaugson D S1010a	Z78524	JF796951	JF797069	JF796890	JF797041
*C. franchetii*	Li JH S6261 (NOCC)	JF796908	JF796953	JF797071	JF796875	JF797024
*C. froschii*	Gunnlaugson D S1150a	JF796909	JF796954	JF797072	JF796876	JF797025
*C. guttatum*	Li JH S6620 (NOCC)	Z78526	JF796955	JF797073	JF796894	JF797049
*C. henryi*	Li JH S6157 (NOCC)	JF796910	JF796956	JF797074	JF796861	JF797011
*C. himalaicum*	-	Z78523	-	-	-	-
*C. irapeanum*	Salazar GA 7234 (MEXU)	FR720328	FR851212	FR851227	FR851217	-
*C. kentuckiense*	Gunnlaugson D S1146a	JF796911	JF796958	JF797076	JF796870	JF797019
*C. lentiginosum*	Li JH S7704 (NOCC)	JF796912	JF796959	JF797077	JF796855	JF797000
*C. lichiangense*	Li JH S6303 (NOCC)	Z78529	JF796960	JF797078	-	JF797001
*C. macranthos*	Ling CY S9032a (PE)	Z78522	JF796962	JF797080	JF796877	JF797026
*C. molle*	Salazar GA 6883 (MEXU)	FR720327	FR851211	FR851226	FR851216	-
*C. montanum*	Frystak M S0613 (PE)	EF370091	JF796965	JF797083	JF796867	JF797017
*C. parviflorum*	Taylor SL S0516 (PE)	EF370089	JF796968	JF797086	JF796863	JF797013
*C. plectrochilum*	Li JH S6301 (NOCC)	Z78528	JF796972	JF797090	JF796891	JF797042
*C. reginae*	Steele B S1159a	JF796917	JF796971	JF797089	JF796860	JF797010
*C. rubitinctum*	Gunnlaugson D S1152a	JF796918	JF796973	JF797091	JF796879	JF797028
*C. segawae*	Gunnlaugson D S1145a	Z78520	JF796974	JF797092	JF796862	JF797012
*C. shanxiense*	Ling CY S0609 (PE)	JF796919	JF796975	JF797093	JF796868	JF797018
*C. sichuanense*	Li JH S6154 (NOCC)	JF796920	JF796976	JF797094	JF796854	JF796999
*C. subtropicum*	Jin XH N8003 (PE)	JF796896	JF796934	JF797053	-	JF796994
*C. tibeticum*	Li JH S6132 (NOCC)	JF796922	JF796978	JF797096	JF796881	JF797030
*C. wardii*	Li JH S6101 (NOCC)	JF796924	-	JF797098	JF796853	JF796996
*C. wumengense*	Li JH S6102 (NOCC)	JF796913	JF796961	JF797079	-	JF797033
*C. yunnanense*	Li JH S6601 (NOCC)	JF796925	-	JF797099	JF796883	JF797032
Outgroup						
*Mexipedium xerophyticum*	Salazar GA 5627	FR720330	FR851210	FR851225	FR851215	-
*Paphiopedilum malipoense*	Liu ZJ E7030a	Z78498	JF796981	JF797101	JF796885	JF797035
*Phragmipedium exstaminodium*	Salazar GA 7499 (MEXU)	FR720329	FR851209	FR851224	FR851214	-
*Phragmipedium schlimii*	Perner H S7032a	Z78514	JF796980	JF797100	JF796884	JF797034

**Table 2 plants-14-00772-t002:** Morphological comparisons and flowering period of W1, *C.* × *ventricosum*, *C. calceolus*, W2, W3, and W4.

Characteristics	*C.* × *ventricosum*	W1	*C. calceolus*	W2	W3	W4
Stem	puberulous	puberulous	glandular-hairy	puberulous	puberulous	puberulous
Leaves	5–6, elliptical or ovate-elliptical, glabrous or occasionally puberulent on the veins on both sides.	5, broadly elliptic to ovate-elliptic, puberulous	5, elliptical or ovate-elliptical, and sparsely pubescent on the back.	5–6, elliptical, glabrous, or occasionally puberulent on the veins on both sides.	5–6, ovate-elliptical, occasionally puberulent on the veins on both sides.	5–6, elliptical or ovate-elliptical, glabrous or occasionally puberulent on the veins on both sides.
	14.0–23.5 × 6.5–11.1 cm	14.0–16.3 × 4.7–7.5 cm	13.2–14.2 × 5.8–6.2 cm	18.0–19.0 × 8.0–9.2 cm	13.0–15.5 × 4.7–7.0 cm	14.2–17.8 × 6.0–7.7 cm
Inflorescences	1- or 2-flowered	1- or 2-flowered	1	1	1- or 2-flowered	1- or 2-flowered
Flowers	purple-red to white	dark reddish-brown to yellow-green	maroon sepals and petals and yellow lip	yellow-green throughout	white to deep pink	light pink throughout
Dorsal sepal	ovate-lanceolate, apex twisted and acuminate	ovate, apex twisted, and acuminate	ovate or ovate-lanceolate, apex twisted and acuminate	ovate, apex twisted, and acuminate	ovate, apex twisted, and acuminate	ovate or ovate-lanceolate, apex twisted and acuminate
	4.4–5.9 × 2.1–2.8 cm	3.9–4.8 × 2.0–2.3 cm	4.4–4.6 × 1.6–1.7 cm	6.5–6.6 × 2.7–3.3 cm	5.4–5.5 × 2.3–2.4 cm	5.0–5.2 × 2.6–3.2 cm
Synsepal	small than dorsal sepal	similar to dorsal sepal, apex shallowly bilobed	similar to dorsal sepal, apex slightly bilobed	small than dorsal sepal	more elongated and thinner than dorsal sepal	small than dorsal sepal
Petals	liner-lanceolate or narrowly ovate-lanceolate, slightly twisted	liner-lanceolate, twisted	linear-tapering or linear-lanceolate, twisted	liner-lanceolate or narrowly ovate-lanceolate, slightly twisted	liner-lanceolate or narrowly ovate-lanceolate, slightly twisted	lanceolate with a relatively smooth margin
	4.3–5.7 × 0.7–1.9 cm	3.6–4.6 × 0.6–0.8 cm	3.7–4.5 × 0.5–0.6 cm	6.3–6.7 × 1.1–1.3 cm	5.8–7.1 × 1.0–1.4 cm	4.3–5.6 × 1.1–1.8 cm
Lip	deeply pouched, subellipsoid, or obvoid-globose, with a paler-rimmed mouth	deeply pouched, ellipsoid, or obovoid-globose, with a yellow-green rim to the mouth	deeply pouched, ellipsoid	deeply pouched, obovoid-globose	deeply pouched, ellipsoid, or obovoid-globose	deeply pouched, ellipsoid, or obovoid-globose
	3.4–3.9 × 2.5–2.9 cm	2.8–3.3 × 1.7–2.7 cm	3.2–3.4 × 2.1–2.2 cm	3.4–3.6 × 2.1–2.3 cm	3.1–3.3 × 2.2–2.3 cm	3.5–3.7 × 2.4–2.7 cm
Stamonode	subovate-oblong	subovate-oblong	suboblong-elliptic	subovate-oblong	subovate-oblong	subovate-oblong
	1.2–1.5 × 0.7–0.9 cm	0.9–1.2 × 0.6–0.7 cm	0.7–1.0 × 0.5–0.7 cm	1.0–1.2 × 0.7–0.8 cm	0.9–1.1 × 0.6–0.7 cm	0.8–1.3 × 0.8–1.0 cm
Flowering period	June	June	June	June	June	June

**Table 3 plants-14-00772-t003:** The flower parts color of W1, *C.* × *ventricosum*, *C. calceolus*, W2, W3, and W4.

Composition	*C.* × *ventricosum*	W1	*C. calceolus*	W2	W3	W4
Lip	Pink White A	Brilliant Greenish Yellow A	Brilliant Greenish Yellow A	Greenish Yellow A	Strong Pink Red A	Pink White A
	Strong Pink Red A	Greyish Brown A	-	-	White A	-
Dorsal sepal	Pink White A	Dark Red A	Dark Red A	Greenish Yellow A	Strong Pink Red A	Pink White A
	Strong Pink Red A	Brilliant Greenish Yellow A	Brilliant Greenish Yellow A	-	White A	-
Synsepal	Pink White A	Dark Red A	Dark Red A	Greenish Yellow A	Strong Pink Red A	Pink White A
	Strong Pink Red A	Brilliant Greenish Yellow A	Brilliant Greenish Yellow A	-	White A	-
Left Petal	Strong Pink Red A	Dark Red A	Dark Red A	Greenish Yellow A	Pink White A to White A	Pink White A
Right Petal	Strong Pink Red A	Dark Red A	Dark Red A	Greenish Yellow A	Pink White A to White A	Pink White A

**Table 4 plants-14-00772-t004:** List of genes in the chloroplast genome of W1.

Category of Genes	Gene Grouping	Name of Genes
Photosynthesis	Subunits of ATP synthase	*atpA*, *atpB*, *atpE*, *atpF* *, *atpH*, *atpI*
	Subunits of NADH dehydrogenase	*ndhA* *, *ndhB* * (×2), *ndhC*, *ndhD*, *ndhE*, *ndhF*, *ndhG*, *ndhH*, *ndhI*, *ndhJ*, *ndhK*
	Subunits of cytochrome	*petA*, *petB* *, *petD* *, *petG*, *petL*, *petN*
	Subunits of photosystem I	*psaA*, *psaB*, *psaC*, *psaI*, *psaJ*
	Subunits of photosystem II	*psbA*, *psbB*, *psbC*, *psbD*, *psbE*, *psbF*, *psbI*, *psbJ*, *psbK*, *psbL*, *psbM*, *psbN*, *psbT*, *psbZ*
	Subunit of rubisco	*rbcL*
Self-replication	Large subunit of ribosome	*rpl14*, *rpl16* *, *rpl2* * (×2), *rpl20*, *rpl22*, *rpl23*(×2), *rpl32*, *rpl33*, *rpl36*
	DNA-dependent RNA polymerase	*rpoA*, *rpoB*, *rpoC1* *, *rpoC2*
	Small subunit of ribosome	*rps11*, *rps12* * (×2), *rps14*, *rps15*, *rps18*, *rps19*(×2), *rps2*, *rps3*, *rps4*, *rps7*(×2), *rps8*
	rRNA	*rrn16*(×2), *rrn23*(×2), *rrn4.5*(×2), *rrn5*(×2)
	tRNA	*trnA-UGC* * (×2), *trnC-GCA*, *trnD-GUC*, *trnE-UUC*, *trnF-GAA*, *trnM-CAU* *trnH-GUG* (×2), *trnI-CAU* (×2), *trnL-CAA* (×2), *trnL-UAG*, *trnL-UAA* *****, *trnV-GAC* (×2), *trnI-GAU* * (×2), *trnR-ACG* (×2), *trnR-UCU*, *trnN-GUU* (×2), *trnfM-CAU*, *trnG-GCC*, *trnG-UCC*, *trnY-GUA*, *trnT-GGU*, *trnT-UGU*, *trnS-UGA*, *trnS-GGA*, *trnS-GCU*, *trnV-UAC* *, *trnW-CCA trnP-UGG*, *trnQ-UUG*
Other genes	Subunitof Acetyl-CoA-carboxylase	*accD*
	c-type cytochrome synthesis gene	*ccsA*
	Protease clpP	*clpP* **
	Translational initiation factor	*infA*
	Maturase	*matK*
Unknown	Conserved open reading frames	*ycf1*, *ycf2*(×2), *ycf3* **, *ycf4*

Note: * gene with a single intron; ** gene with 2 introns; (×2) duplicated gene.

**Table 5 plants-14-00772-t005:** RSCU analysis of amino acids in the chloroplast genome of W1.

AminoAcid	Codon	Number	RSCU	Amino Acid	Codon	Number	RSCU
Phe	UUU	814	1.32	Pro	CCU	331	1.54
	UUC	422	0.68		CCC	197	0.92
Ser	UCU	470	1.71		CCA	244	1.13
	UCC	283	1.03		CCG	88	0.41
	UCA	328	1.20	His	CAU	397	1.52
	UCG	132	0.48		CAC	125	0.48
Tyr	UAU	652	1.62	Gln	CAA	585	1.53
	UAC	154	0.38		CAG	182	0.47
Cys	UGU	185	1.47	Arg	CGU	277	1.34
	UGC	67	0.53		CGC	78	0.38
Leu	UUA	698	1.91		CGA	282	1.36
	UUG	462	1.26		CGG	88	0.43
	CUU	449	1.23	Ile	AUU	881	1.47
	CUC	145	0.40		AUC	357	0.59
	CUA	297	0.81		AUA	566	0.94
	CUG	145	0.40	Met	AUG	512	1.00
Trp	UGG	378	1.00	Thr	ACU	429	1.63
Asn	AAU	804	1.60		ACC	187	0.71
	AAC	203	0.40		ACA	325	1.23
Lys	AAA	836	1.47		ACG	112	0.43
	AAG	299	0.53	Val	GUU	417	1.45
Ser	AGU	335	1.22		GUC	134	0.46
	AGC	98	0.36		GUA	430	1.49
Arg	AGA	390	1.89		GUG	172	0.60
	AGG	126	0.61	Ala	GCU	515	1.81
Asp	GAU	712	1.62		GCC	156	0.55
	GAC	165	0.38		GCA	349	1.23
Glu	GAA	871	1.48		GCG	118	0.41
	GAG	309	0.52	Gly	GGU	438	1.21
TER	UAA	24	1.38		GGC	156	0.43
	UAG	15	0.87		GGA	591	1.64
	UGA	13	0.75		GGG	260	0.72

**Table 6 plants-14-00772-t006:** Correlation analysis of GC content and ENC value at each base position of the chloroplast genome codon of W1.

Index	GC1	GC2	GC3	GCall	ENC
GC1	1				
GC2	0.397 **	1			
GC3	0.204	0.094	1		
GCall	0.824 **	0.734 **	0.481 **	1	
ENC	0.182	−0.216	0.545 **	0.177	1

Note: ** indicates a significant correlation (*p* < 0.01).

**Table 7 plants-14-00772-t007:** Statistics from phylogenetic analysis.

DNA Region	No. of Taxa	Aligned Length (bp)	No. Variable Characters (%)	No. Informative Characters (%)	Consistency Index	Retention Index
nrITS	42	803	496 (61.77)	320 (39.85)	0.635	0.784
plastid	42	3679	1293 (35.15)	828 (22.51)	0.824	0.916
combine	42	4482	1789 (39.92)	1148 (25.61)	0.674	0.814

## Data Availability

The original contributions presented in the study are included in the article, further inquiries can be directed to the corresponding authors.

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
