# Peer review of "Unveiling a Meaningful Form of Cypripedium × ventricosum Sw. (Cypripedioideae, Orchidaceae) from Changbai Mountain, China: Insights from Morphological, Molecular, and Plastome Analyses"

_plants, 2025, doi:10.3390/plants14050772_

Round 1

Reviewer 1 Report (New Reviewer)

Comments and Suggestions for Authors

Dear authors

I find the paper interesting and appropriate with extensive experimental basis.

It in principle merits, in my oppinion, to be pubished but it bears the burden of an heterogeneous English quality that sometimes is very low and extraordinarily difficults understanding the meaning of entire paragraphs. This is particularly perceived in the Abstract, introduction and part of the botanical text.

I proposed some improvements in the attached pdf for determined paragraphs. But I strongly advice you to entirely revise the English text in order to improve the readability, academic tone and flow thorough your text.

Concerning the publication of the new nothospecies, please read carefully the International Code for Nomenclature of Plants, Algae and Fungi and follow closely the precepts there in order to validly publish your new taxon.

Adding one figure with the image, good resolution, of the Holotype herbarium sheet, would be highly useful

Comments on the Quality of English Language

The quality is heterogeneous since some parts, in particular those concerning molecular evidence are almost acceptable, while botanical, introductory and other texts sucha as the abstract require extensive edition. 

Author Response

Reviewer 2 Report (New Reviewer)

Comments and Suggestions for Authors

The manuscript is relevant and in line with the scope of the journal. The quality of the drafting is quite good, but there are some significant shortcomings that need to be addressed.

1. I recommend changing the title of the article to one that is clearer, more precise and more in line with the content of the article.

2. I strongly doubt the claim that this study can and should be used to determine the position of a new hybrid in the evolution of the genus Cypripedium (line 98). The questions to be addressed should be formulated in a way that is consistent with the content of the paper and should exclude what is not really achievable by this study. 

3. The terminology in the article needs to be very strict. A hybrid that is discovered for the first time cannot really be called a species. It is a new hybrid of species rank and should be consistently referred to as a hybrid. 

4. Which three species were sampled with 1 g of leaves each (line 108)? They must be clearly identified and consistency must be maintained throughout the text.

5. Why is it stated (line 123) that the leaves were frozen, when it was previously stated (line 110) that the leaves were dried in silica gel? The methodology needs to be clear and consistent so that there is no doubt.

6. I strongly recommend that the description of the new nothotaxon be moved to the beginning of the results and that the novelty of the taxon be fully substantiated. This would avoid ambiguities in the application of nomenclature rules. At present, a new nothotaxon is mentioned many times before its formal description, characterised in all sorts of ways, and only afterwards its name is validated. 

7. The quality of all the graphs and figures is unsatisfactory (with the possible exception of Figure 2, which is of marginal quality). The quality of Figures 3, 6, 7, 8 and 9 is particularly poor, as most of the entries are illegible. Figure 3 in the manuscript submitted for assessment is of such poor sharpness that it can only be seen as a colour patch.

8. I strongly recommend that the captions of the tables be expanded to explain the full content of the tables, rather than devoting half of the results paragraph to describing the content of the table (Table 7, 3.6 subchapter), with hardly any results. Tables should complement the text but not vice versa. 

9. The unnumbered subsection "Taxonomy" should be moved to the results section. In fact, that part is not only taxonomy but also nomenclature, so it would be more accurate to call it "New nothotaxon". 

10. The description of the new hybrid does not meet all the requirements for describing a new taxon. I strongly recommend that you consult the Code (https://www.iapt-taxon.org/nomen/main.php), paying particular attention to Chapter H. Failure to comply with the formal requirements of the Code may result in the description of the new taxon being regarded as not valid. 

11. After the name of the new taxon of hybrid origin in the protologue (line 398), it must be stated that it is a new nothotaxon (Cypripedium × wangqingensis L.F. Chen, nothosp. nov.) and the putative or defined parental species must be indicated (See Codex (https://www.iapt-taxon.org/nomen/main.php; Chapter H). 

Comments on the Quality of English Language

Terminology must be revised and corrected.

Author Response

Reviewer 3 Report (New Reviewer)

Comments and Suggestions for Authors

This is a not very novel, but useful paper.  Natural hybrids identification is the basis for the study of plant evolution and phylogeny. Therefore, even if  neither the  methodology,  nor the results found  are really novel, the paper is  worth to be published.

I had no real problems with the way how the data were  handed and analyzed. The design of the paper is rather complicated come out but this is the fate of papers dealing with this kind of analysis.

So come on in a sense, I do not have any critical comments for things to be improved.

Round 2

Reviewer 1 Report (New Reviewer)

Comments and Suggestions for Authors

You did an extraordinary task in improving the manuscript. Congratulations!

The image of the holotype is highly appropriate

I have minor obbservations which can be found in the attached pdf file.

I suggest to do not delete the drawing, which supplies detailed information and complements the photographs. I propose to cite it in the discussion as Figure 11 and include the appropriate figure caption, including the authorship of this nice illustration

Comments on the Quality of English Language

The English quality is acceptable and highly improved in contrast with the previous version. However minor problems persist. I marked those I detected and suggested minor changes

I suggest you to work with a clean version and read it again to trace remaining issues.

Author Response

Reviewer 2 Report (New Reviewer)

Comments and Suggestions for Authors

Since the authors have clarified the questions asked after the first round of reviews and have added to the manuscript, significant new considerations have arisen for the description of the new species. As the authors have identified the parental species, the new hybrid described cannot be considered a new hybrid as it represents a hybrid swarm.    Article H.4.1 (attached) of the Code (https://www.iapt-taxon.org/nomen/main.php) explicitly states that the nothotaxon includes not only F1 generation, but all offspring and backcrosses with parental species of subsequent generations.This makes the description of a new hybrid meaningless.The authors must first solve this major problem.   I agree that the article is worthy of attention, but nomenclatural issues cannot be ignored and the whole concept of the manuscript needs to be changed. Thus, according to the rules of nomenclature, the nothotaxon described will become a synonym of Cypripedium × ventricosum Sw.   H.4.1. When all the parent taxa can be postulated or are known, a nothotaxon is circumscribed so as to include all individuals recognizably derived from the crossing of representatives of the stated parent taxa (i.e. not only the F1 but subsequent filial generations and also back-crosses and combinations of these). There can thus be only one correct name corresponding to a particular hybrid formula; this is the earliest legitimate name (Art. 6.5) at the appropriate rank (Art. H.5), and other names corresponding to the same hybrid formula are synonyms of it (but see Art. 52 Note 4).

Round 3

Reviewer 2 Report (New Reviewer)

Comments and Suggestions for Authors

The updated and restructured version of the manuscript is a significant improvement and, in my opinion, can be published with minor editorial corrections. A few comments on the content: 

  1. The English term form (taxonomic rank, Latin forma) and the term form, which defines variation but does not denote taxonomic rank, should be clearly distinguished in the text. The Latin term forma (instead of the English form) now used in the manuscript gives the impression that the authors are talking about taxonomic rank. I therefore suggest that the English term (form, plural forms) be used consistently throughout the text instead of the Latin term (forma, plural formae).
  2. The abbreviation "sp." is used incorrectly (= species), because the article is not referring to species in the classical sense, but to individuals resulting from introgressive hybridisation. In many cases the abbreviation is superfluous and only leads to taxonomic confusion. The generic name Cypripedium would be perfectly adequate, and if necessary, other means (plants, individuals, hybrids, etc.) could be used to clarify the sentence. Instead of sp.1, sp.2, etc., I recommend the use of non-confusing abbreviations such as H1, H2 or any other abbreviation of your choice.

  3. The graph in Figure 5 needs to be significantly enlarged as it is now completely illegible. What does Cypripedium sp. mean in this case? It needs to be very clearly and precisely stated throughout so that there is no confusion and no guesswork as to what is being said. It is now clear that the graphic shows the genome of some unknown Cypripedium.

  4. The same remark applies to the caption of Table 4.

Round 4

Reviewer 2 Report (New Reviewer)

Comments and Suggestions for Authors

The authors have taken due account of all comments. I have not noticed any major flaws in the manuscript. The remaining minor technical shortcomings can easily be corrected in the final stages of editing and proof reading.

This manuscript is a resubmission of an earlier submission. The following is a list of the peer review reports and author responses from that submission.

Round 1

Reviewer 1 Report

Comments and Suggestions for Authors

Row 44: Orchids write in lowercase letters throughout the manuscript.

Row 50: Change Pftzer to Pfitzer. Also, in reference list change.

Row 51: Term „taxonomically expanded” replace with some other term or try to rephrase sentence e.g. “According to Pfitzer, genus Cypripedium comprises 28 species included in four sections.”

Rows 58 and 59: Rephrase this part of sentence “… has greatly advanced our understanding of Orchid relationships, clarifying relationships among Orchid subfamilies with morphologically confused taxa.” There is no need to use relationships twice in the same sentence.

e.g. “has clarified and considerably advanced our understanding of the relationships between orchid subfamilies with morphologically similar taxa.”

Row 61: Change “low copy” with “low-copy”.

Row 93: “thus confirmed a natural hybrid”. In my opinion this should be in Disscution section. In this part you should elaborate why did you performed your study, not to put conclusions in introduction section. Based on the morphology, it is obvious that you found a hybrid and that it is similar with C. calceolus and C. × ventricosum. However, the analyzes are there to confirm it and according to the conclusions, the final name and description presented.

Row 97: Add “Jilin Provice, Yanbian Chaoxianzu (Korean) Autonomous Prefecture”.

Row 98: I would avoid using word “population” in this context. In my opinion, it is not one population since we regard C. × ventricosum and C. calceolus as separate taxa that form hybrid taxon Cypripedium × “wangqingensis”. Perhaps it should be rephrased to mean a locality where taxa grow naturally together.

Row 100: Change “known” into “named”. I suggest to wangqingensis put in quotation marks since it was provisional or as you wrote temporarily name in the beginning of study. After your analysis you give conclusions and explanation that it is a new taxon.

Figure 1. Caption change „provice“ to „province“. If it is explained in figure caption what is A and B then there is no need to be writen also on the figure under A and B. Instead of the map of Jilin Province, I suggest inserting a small map of China with the provinces where Jilin Province would be marked with different colour. I also suggest adding a point on map B to show the exact location of the locality under investigation.

Row 106: How did you carry out the measurements of the morphological characteristics given in Table 2? Did you measured leaves and sepals with a ruler in the field or using herbarium specimens, etc. If you took measurments from photographs, explain how you did that.

Table 1. It is not clear what all this information in the Vouchers column means. Are they herbarium numbers, legator names, herbarium acronyms according to the Index Herbariorum? Explain that in heading of the table or cite studies where those sequences were published.

Table 2. In section „Synsepal” change “speal” to “sepal”. Table 2 also shows the flowering period of investigated taxa, so include this in the heading of the table since flowering period is phenology not a morphological characteristic. I suggest that you reorder table so that data of hybrid C. × wangqingensis are in the middle.

Figure 2. I suggest to put picture A between C and D. Also, instead of using apostrophe - S, I suggest to put in brackets is it flower – front or latteral view etc.

Figure 3. I would put figure 3 after description of Cypripedium × wangqingensis. It is not clear from the illustration under B what it is. It is not a fruiting plant, so rephrase it in the caption to be more specific. My guess is that these sepals are in the fruiting stage.

Row 201: Avoid “concluded”. This is the Result section, where you simply present your results. In the Discussion section, you then present and explain your results and conclusions.

Row 202: “We compared flower of C. × wangqingensis with RHS Colour Chart (Table 3).” This is already described in Morphological analysis in the Material section. Here under Results you can write that the colour of the different flower parts is given according to the RHS Colour Chart in Table 3.

Table 3. I suggest that you give colors of same flower parts of C. calceolus and C. × ventricosum.

Row 330: Taxonomy should be in section Disscution. After analyzes and resultes you prove and disscus that it is natural hybrid and as a part of conclusion you give description of new hybrid.

Comments on the Quality of English Language

I suggest English checking.

Reviewer 2 Report

Comments and Suggestions for Authors

16-18 means what?

40. Mountains protected?

44. Complete phylogeny of Orchidaceae? We aren't close...

47. Minority scholar?

54. Multiple species. ?8.

Fig. 1 not very helpful - Where is Jilin Province, (adjacant provinces, countries?) collection site could be approximately indicated.

107. Rephrase.

132. Call a plastome a plastome, not genome.

143, 256. Explain abbreviations (ENC, CDS)

151-156?

Table 1. I presume there will be accession numbers for C. X wangqingensis.

187-189.?

Fig 2. Confusing, Make sure letters are close to part of plant. Does F refer to the image showing the parts of the plant? If not, what does?

Fig. 3. Fruiting plant is where> Reorganize caption.

Table 2 and thereabouts. speals, collum, twisted weeks, etc.???? Note the dorsal sepal apex can be both twisted and acuminate - clarify variation here. The rim colour of C. calceolus should be specifically mentioned.

335. Rewrite, some colours in capitals, etc..

354. All species should be cited in the same way.

364/365. "Fredy L's" hypothesis - surely not unique; cite this paper properly.

Comments on the Quality of English Language

I am afraid that the paper must be completely rewritten.